# Rice Biofortification: High Iron, Zinc, and Vitamin-A to Fight against "Hidden Hunger"

**Shuvobrata Majumder**, **Karabi Datta and Swapan Kumar Datta ***

Laboratory of Translational Research on Transgenic Crops, University of Calcutta, Annex Building II,
35 Ballygunge Circular Road, Kolkata 700019, India; shuvobratamajumder@yahoo.co.in (S.M.);
krbdatta@yahoo.com (K.D.)
* Correspondence: swpndatta@yahoo.com or skdbot@caluniv.ac.in; Tel.: +91-876-864-4777

**Abstract:** One out of three humans suffer from micronutrient deficiencies called "hidden hunger". Underprivileged people, including preschool children and women, suffer most from deficiency diseases and other health-related issues. Rice (*Oryza sativa*), a staple food, is their source of nutrients, contributing up to 70% of daily calories for more than half of the world's population. Solving "hidden hunger" through rice biofortification would be a sustainable approach for those people who mainly consume rice and have limited access to diversified food. White milled rice grains lose essential nutrients through polishing. Therefore, seed-specific higher accumulation of essential nutrients is a necessity. Through the method of biofortification (via genetic engineering/molecular breeding), significant increases in iron and zinc with other essential minerals and provitamin-A (β-carotene) was achieved in rice grain. Many indica and japonica rice cultivars have been biofortified worldwide, being popularly known as 'high iron rice', 'low phytate rice', 'high zinc rice', and 'high carotenoid rice' (golden rice) varieties. Market availability of such varieties could reduce "hidden hunger", and a large population of the world could be cured from iron deficiency anemia (IDA), zinc deficiency, and vitamin-A deficiency (VAD). In this review, different approaches of rice biofortification with their outcomes have been elaborated and discussed. Future strategies of nutrition improvement using genome editing (CRISPR/Cas9) and the need of policy support have been highlighted.

**Keywords:** hidden hunger; biofortification; nutritional security; biofortified rice; high iron rice; ferritin rice; low phytate rice; high zinc rice; golden rice

## 1. Introduction

A quick meal may satisfy hunger, but there is a deeper problem of "hidden hunger" which is only fulfilled by nutritionally enriched food. Having a balanced diet is a far-fetched dream for the underprivileged people of the world. A carbohydrate-rich diet including rice, wheat, or maize (the major staple food) is consumed worldwide and mainly contributes to solving the problem of hunger, however, "hidden hunger" still persists in the world. "Hidden hunger" is caused when the body is deprived of essential micronutrients. It remains hidden or unnoticed and only surfaces when a deficiency symptom is diagnosed. Nutrient deficiency or malnutrition has affected at least 2 billion people (or 1 out of 3), mostly in Africa, South Asia, and Latin America [1]. Micronutrient deficiency is a silent epidemic condition—it slowly weakens the immune system, stunts physical and intellectual growth, and even causes death. Under micronutrient deficiencies, iron deficiency or iron deficiency anemia (IDA), zinc deficiency, and vitamin-A deficiency (VAD) are widespread and cause serious consequences. More than 24,000 people globally die daily owing to "hidden hunger" and malnutrition [2]. To combat these deficiencies, fortification of food with different biological and chemical supplements and the alternation of the food processing system are essential. Biofortified

(including bioengineered) staple food crops is a sustainable alternative that can be highly beneficial for people who have limited access to varied dietary resources.

Genetically modified (GM) rice specifically developed to fight against "hidden hunger" is the most promising over any other staple crops because half of the world's population depend on rice. Hulling of field harvested paddy (rough rice) produces brown rice, the most nutritious form of processed rice. Unpolished brown rice contains important minerals such as iron, zinc, copper, calcium, phosphorus, and vitamins such as thiamin ($B_1$), riboflavin ($B_2$), niacin ($B_3$), pantothenic acid ($B_5$), pyridoxine ($B_6$), biotin ($B_7$), folate ($B_9$), and $\alpha$-tocopherol (E), but does not contain vitamins A, D, or C [3]. However, the average consumer prefers white rice grains with lightness, softness, easy digestibility, better eating characteristics, and shorter cooking time. White polished (milled) rice loses the bran layer along with subaleurone, embryo, and a small part of the endosperm underneath [4]. Polished (milled) rice is lower in nutritional quality than brown rice (Figure 1), as its iron content is reduced by 2.14 times (from 8.8 to 4.1 ppm) to 4.75 (from 19 to 4 ppm), zinc by 1.83 times (33 to 18 ppm), along with other minerals, vitamins, fats, proteins, and fibers [5–7]. Nonetheless, these amounts of reduction in minerals may vary among the rice cultivars and the grain milling processes. Education and awareness has increased brown rice consumption, yet a vast majority of rice consumers still prefer white polished rice, therefore leaving it to the scientists to consider developing nutritionally enhanced rice varieties through biofortification (endosperm specific) that remain nutritious even after processing and polishing.

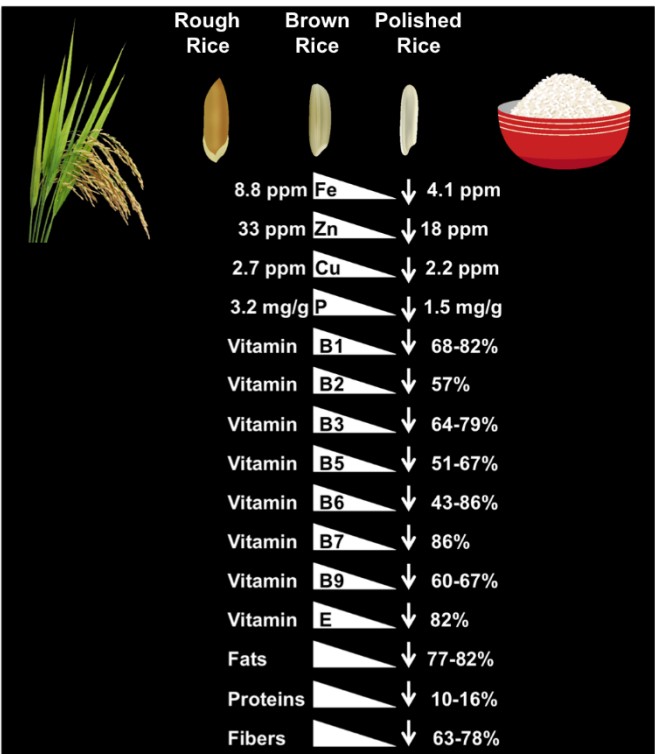

**Figure 1.** Loss of nutrients and minerals from rice grain due to milling process. (From left to right) Mature rice ready for harvesting, paddy (rough rice), brown rice, polished (milled) rice, cooked rice ready for consumption. This info-graphic has been made from the information provided in [5–7].

## 2. Rice Biofortification

Biofortification is considered to be an effective process to increase the micronutrients in food crops including rice. It is also a sustainable and feasible strategy to alleviate micronutrient deficiencies for people who mainly consume rice and have limited access to diversified food (or food markets) and good health facilities [8]. Under rice biofortification research projects for maintaining, increasing, and introducing new micronutrients in rice grain, different approaches have been strategized worldwide.

Such significant and successful approaches of rice biofortification in three broad areas are discussed in this article as follows:

- High iron rice;
- High zinc rice;
- Golden rice (high carotenoid rice).

## 3. High Iron Rice

Iron is one of the essential minerals for human health. Rice iron concentration becomes drastically reduced more than any other mineral due to post-harvest processing. Paddy (rough rice) contains 38 ppm of iron that is reduced to 8.8 ppm in brown rice after processing and finally 4.1 ppm in milled rice [6] (Figure 1). In another report, iron concentration in brown rice that was 19 ppm became reduced to around 4 ppm in polished grains (a reduction of 4.75 times) [7]. This evident reduction of iron in consumable rice grain is the concern that gave rise to iron biofortification specific to milled rice. Availability of adequate iron in rice would help to maintain the health of children and pregnant women in developing countries. Deficiency of iron causes IDA that has serious consequences on human health, specifically on children and women. IDA affects 32.9% of the world population, with the risk being higher in Saharan Africa and South Asian countries [9]. It causes impaired cognitive development in children, weakens the immune system, and increases the risk of morbidity. IDA can also adversely affect productivity, cause premature births, and increase the risk of mortality in women. According to WHO's Nutrition Landscape Information System (NLiS), data of the 10 most populated Asian countries indicate that Pakistan had the highest anemic children (61%) in 2011 and India had the highest number of anemic pregnant women (51.5%) in 2016 (Figure 2). Development of high iron milled rice under a biofortification project could be effective against IDA in such affected countries.

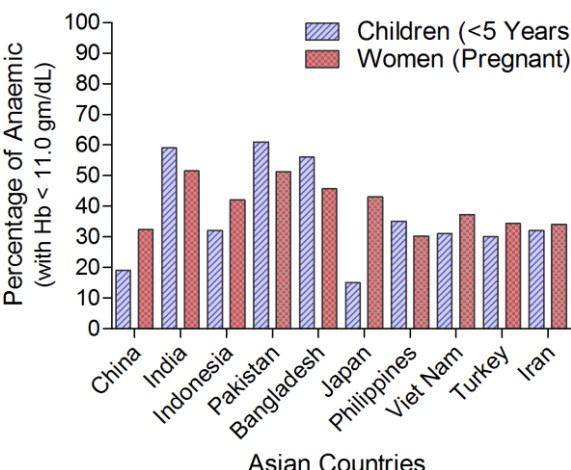

**Figure 2.** Ten population country-wise percentages of anemic children and pregnant women (Nutrition Landscape Information System (NLiS), WHO).

### 3.1. Iron Biofortification via Conventional Plant Breeding

Iron biofortification via conventional plant breeding has always been of interest to plant breeders to create a high iron rice varieties with high yield, disease tolerance, and quality seed vigor. One such rice variety, the IR68144, was developed by crossing between IR8 and Taichung (Native)-1 [10]. The IR68144 is semi-dwarf and contains high yield properties, producing 21 μg/g of iron concentration in brown rice and retaining about 80% of its iron concentration even after polishing when compared to other control varieties [11]. Tests on Filipino women (nonanemic) confirmed that consumption of the IR68144 rice variety was an improvement in terms of iron health [12]. This IR68144 rice cultivar further serves as a prime choice of variety for further transgenic approaches [13,14].

### 3.2. Iron Biofortification via Molecular Plant Breeding

Molecular plant breeding is now considered as an efficient and reliable method of studying the genotype–phenotype relationship as compared to conventional breeding. Table 1 shows some popular approaches of iron fortification of rice achieved by different attempts via molecular breeding.

**Table 1.** Approaches for iron biofortification in rice along with fold of iron increment (compared with control rice) in polished rice or in brown rice.

| Rice Iron Biofortification Strategies | Iron Increased in Biofortified Rice Seed | Reference |
|---|---|---|
| Improving iron storage via ferritin | 1.5–2.2-fold in brown rice (BR) <br> 2.0–3.7-fold in polished rice (PR) | [15,16] <br> [13,17–20] |
| Chelation based strategy (via *NAS* gene) | 2.3–4.0-fold in PR | [7,21–23] |
| Enhancing iron influx (via *OsYSL2* gene) | 4.4-fold in PR | [24] |
| Enhancing iron uptake and translocation (via *IDS3* gene) | 1.3-fold in BR <br> 1.4-fold in PR | [25] <br> [26] |
| Enhancing iron translocation (silencing *OsVITs* genes) | 1.4-fold in BR <br> 1.8-fold in PR | [27] <br> [28] |
| Manipulation of iron uptake and translocation regulators | 2.0–3.8-fold in BR <br> 2.9-fold in PR | [29,30] <br> [29] |
| Low phytate rice (RNAi silencing of phytic acid) | 1.3–1.8-fold in PR | [31–33] |
| Release of phytic acid bound iron (by *phytase* gene) | 2.0-fold in BR <br> 2.0–6.3-fold in PR | [34] <br> [35–37] |
| Multiple transgenes combination | 6.0-fold in BR <br> 3.4–6.0-fold in PR | [38] <br> [39–42] |

#### 3.2.1. Enhancement of Iron Storage in Rice

In most organisms, including plants, there is an iron storage protein—Ferritin [43]. Plant ferritin is a large protein with 24 subunits, which has ferroxidase activity and is capable of storing up to 4500 iron atoms in a nontoxic complex form [44,45]. The *ferritin* genes of many plants have been isolated and sequenced but the soybean *ferritin* has been studied in detail. In soybean, the two types of ferritin proteins that are present are encoded by *SoyferH1* and *SoyferH2* ferritin genes [46]. The human intestine can efficiently absorb iron from the soybean ferritin iron complex; therefore, the soybean *ferritin* gene was considered as a candidate gene for iron biofortification in rice [47]. In multiple experiments, endosperm specific promoters were used, specifically the rice globulin (*OsGlb*) promoter and the rice glutelin (*OsGluB1*) promoter for *ferritin* gene expression in rice, resulting in up to 3.7-fold iron increase in rice grain [13,15–20].

High iron rice, developed from molecular breeding, could be used as donor material in subsequent interbreeding programs for high iron local rice variety development. Vasconcelos et al., developed a high iron IR68144 rice variety by overexpressing soybean *ferritin* gene, which increased iron concentration by 3.7-fold in polished rice grain [13]. Paul et al., successfully interbred IR68144 rice with a high-yielding rice cultivar—Swarna [14]. This resulted in a new variety with 2.54-fold more iron and 1.54-fold more zinc in milled rice grain as compared to control Swarna. Such introgressed breeding projects have a positive impact in developing country-wise local (popular) high iron rice varieties and help fight against IDA.

#### 3.2.2. Enhancement of Plant Iron Uptake from the Soil via Chelation-Based Strategy

Under conditions of low iron, rice plants can increase iron uptake from soil through chelation-based strategy, the same as other graminaceous staple crops such as wheat and maize. This strategy transports $Fe^{3+}$ from rhizosphere to plant roots with the help of soluble phytosiderophores (PS). PS, like mugeniec acid (MA) and avenic acid, are small, high-affinity iron-chelating organic compounds secreted by

plants under iron- or zinc-deficient conditions, which can chelate iron or zinc and increase their uptake by plant roots [48,49]. In rice plant, nicotinamine synthase (NAS) and nicotinamine transferase are the main enzymes involved in the release of phytosiderophores with the help of TOM1 (transporter of MAs) transporter [50,51]. Specifically, MA family phytosiderophores have a higher affinity towards $Fe^{3+}$ and play a vital role in rice plants [52]. MAs bind $Fe^{3+}$ efficiently, forming complexes, and transport into the root via *yellow stripe 1* (YS1) transporter [53]. To achieve iron-fortified rice, overexpression of genes involved in MA biosynthesis was considered by many scientists. Attempts were made by overexpressing NAS, as it catalyzes the synthesis of nicotianamine (NA) from S-adenosyl methionine [35]. Three NAS genes have been identified from the rice genome- *OsNAS1*, *OsNAS2*, and *OsNAS3,* and overexpression of these genes gave satisfactory results and rice iron concentration was increased [46]. Overexpression of rice *OsNAS1*, *OsNAS2*, and *OsNAS3* was done by Johnson et al., [21], *OsNAS2* was overexpressed by Lee et al., [22] and *OsNAS3* by Lee et al. [23]. Barley *HvNAS1* gene was expressed in rice by Masuda et al. [7]. Iron concentration in polished rice was reported to be more than double in these attempts.

### 3.2.3. Enhancement of Iron Influx in Seeds

In rice, a total of 18 different yellow stripe-like (*YSL*) genes play an important role as metal (iron)-chelator transporters in endosperm [24,54]. This group of transporters is involved in long-distance transport of iron-NA complex via phloem, and iron influx into rice endosperm was found to be controlled specifically by the *OsYSL2*, an iron nicotianamine transporter [54,55]. The importance of the *OsYSL2* gene was demonstrated in rice plant by Ishimaru et al. [24]. Disruption of the *OsYSL2* gene decreased iron concentration by 18% in brown rice and by 39% in polished rice, as compared to control plants [24]. In another experiment, when the *OsYSL2* gene was overexpressed under the sucrose transporter (*OsSUT1*) promoter in rice, it increased iron concentration by about fourfold in polished grain [24]. This approach of overexpression of *OsSUT1* promoter-driven *OsYSL2* gene was found effective for iron biofortification. In future, combination of other *OsYSL* genes with different promoters could be more effective.

### 3.2.4. Enhancement of Iron Uptake and Translocation

Different types of MA genes from different plant sources have been introduced in rice for the enhancement of iron uptake and translocation. MAs genes found in barley synthesized different types of MA compared with rice MAs, and in an iron-deficient environment activates its iron deficiency specific clone no. 2 (*IDS2*) and no. 3 (*IDS3*), thereby playing an important role in combating iron deficiency [39,56,57]. The *IDS* genes can synthesize special types of MAs through 2'-deoxymugineic acid (DMA). On the contrary, rice lacks the ability to synthesize other types of MAs apart of DMA, whereas barley secretes different types of MAs such as MA, 3-epihydroxymugineic acid (epi-HMA), and 3-epihydroxy-2'-deoxymugineic acid (epi-HDMA) [58]. Barley's $Fe^{3+}$-MA complex has a better stability than rice $Fe^{3+}$-DMA complex in slightly acidic soil [59]. Therefore, expressing barley *IDS* genes in rice by molecular breeding could enhance iron uptake from soil and its translocation in rice plant tissues. In 2008, this approach was implemented by Masuda et al. and Suzuki et al. who developed IDS3 rice lines that were able to increase iron concentration by 1.4-fold and 1.3-fold in both polished and brown grains, respectively, as compared to control [25,26]. With the availability of genome sequencing data of different graminaceous plants, in the near future more types of IDS or IDS-like genes could be identified and could be used for iron biofortification in rice research.

### 3.2.5. Enhancement of Iron Translocation

Rice plants encode different types of metal transporter genes and the products of such genes play an important function in metal translocation throughout the whole plant and grains. The rice vacuolar iron transporter genes (*OsVIT1* and *OsVIT2*) are examples of such types that are specifically involved in transportation of $Zn^{2+}$ and $Fe^{2+}$ into vacuoles via tonoplast [60]. These genes are ubiquitously

expressed in the whole rice plant in low levels but very high expression is found in flag leaves [27]. The knockdown of *OsVIT* genes in rice increases iron and zinc accumulation in the grains and decreases them in the flag leaves [28]. In *OsVIT1* and *OsVIT2* gene knockout rice, there was an increase of 1.4-fold iron in rice grain [27]. One concern of such rice line was that if they were grown in $Cd^{2+}$ polluted soil, accumulation of $Cd^{2+}$ concentration was detected. Hence, further understanding of $Cd^{2+}$ binding domain and its regulatory mechanism is required to prevent toxic (non-essential) metal accumulation in biofortified rice for safe consumption.

### 3.2.6. Manipulation of Iron Uptake and Translocation Regulators

Kobayashi et al. identified two negative regulators of iron deficiency responses in rice, that is, OsHRZ1 and OsHRZ2 ubiquitin ligases [29]. These *Oryza sativa* haemerythrin motif-containing really interesting new gene (RING)- and zinc-finger protein 1 (OsHRZ1) and OsHRZ2 bind with iron and zinc, and possess ubiquitination activity. RNAi-mediated silencing of OsHRZ2 in rice (RNAi-HRZ2) resulted in better iron accumulation as found in shoots and seeds compared to control plants. The RNAi-HRZ2 rice plants contained about 3.8-fold more iron in brown rice and about 2.9-fold more iron in polished rice grain compared to that of control rice (Table 1). Enhanced expression of other iron utilization-related genes was found in RNAi-HRZ2 rice plants.

Iron deficiency stress activates a basic helix-loop-helix (bHLH) transcription factor, OsIRO2, that acts as a positive regulator of iron deficiency responses in rice [61]. Overexpression of OsIRO2 resulted in 2.0-fold higher amounts of iron in brown rice grains of transgenic rice than control rice [30]. The OsIRO2 overexpressed rice plants effectively accumulated iron even when cultivated in calcareous soil and also showed zinc accumulation in grains [30]. This strategy of iron biofortification of rice could prove most effective where rice cultivation is dependent on calcareous soil.

### 3.2.7. Low Phytate Rice by Using RNAi Technology

In most cereals, approximately 80% of the total phytic acid gets accumulated in the aleurone layer of the grains with the exception in maize. Phytic acid accumulates as mixed salts called phytate. Phytate has six negatively charged ions, making it a potent chelator of divalent cations such as $Fe^{2+}$, $Zn^{2+}$, $Ca^{2+}$, and $Mg^{2+}$ and reduces bioavailability of such important divalent minerals (Figure 3).

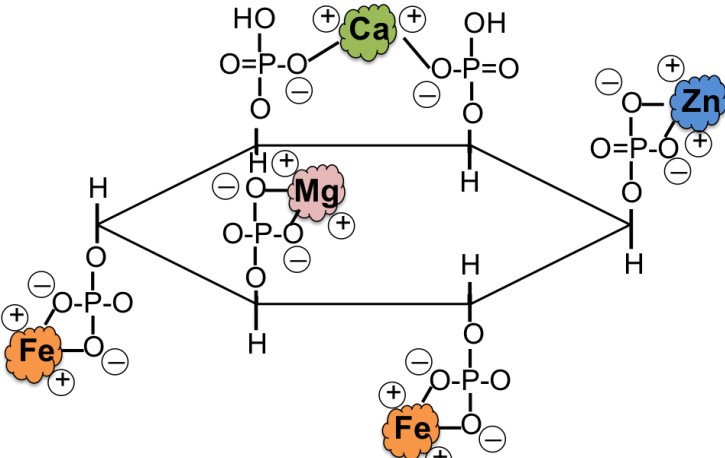

**Figure 3.** Phytic acid as chelator of divalent cations of iron ($Fe^{2+}$), zinc ($Zn^{2+}$), calcium ($Ca^{2+}$), and magnesium ($Mg^{2+}$).

Many attempts have been made to reduce the phytic acid concentration in rice by generating mutant varieties exhibiting a low phytic acid (*lpa*) phenotype [62,63]. Although these mutant lines are effective, they compromised crop yield and overall performance. As an alternative strategy, transgenic crops were developed by manipulating the phytic acid biosynthetic pathway (Figure 4) by RNA interference (RNAi)-mediated silencing its key enzymes [31,64,65].

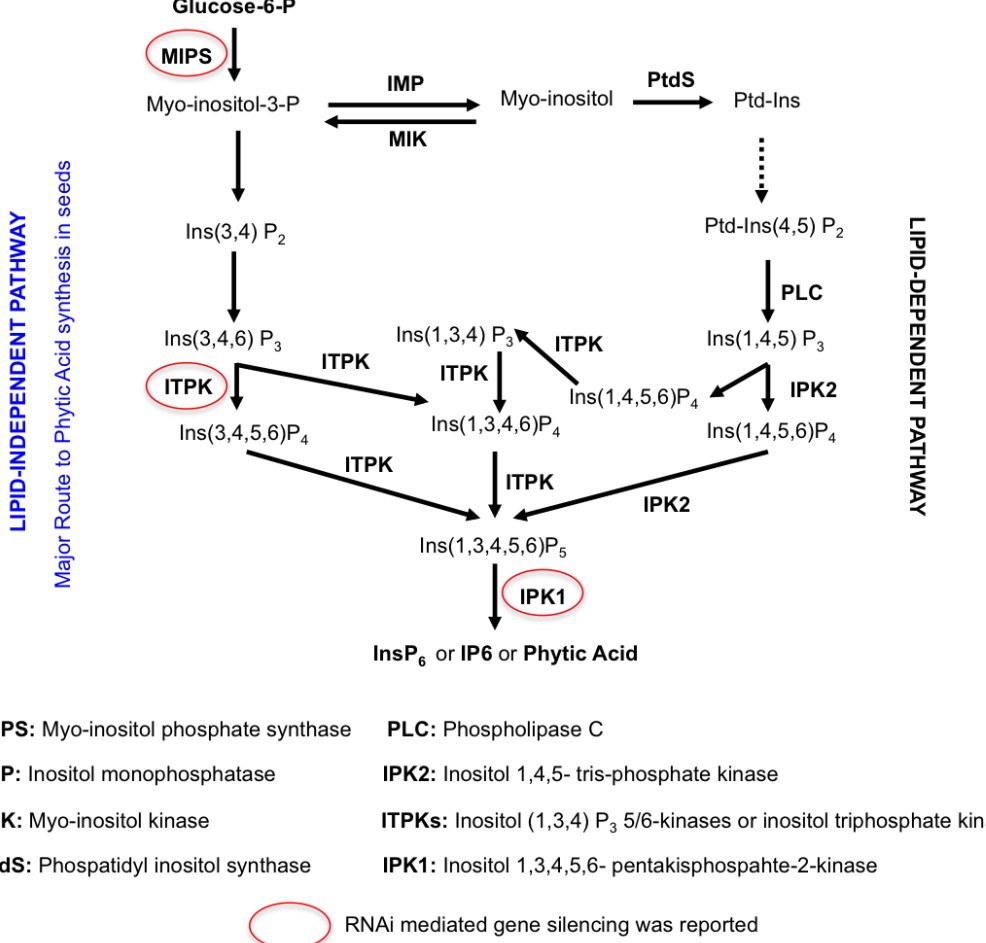

**MIPS:** Myo-inositol phosphate synthase　　**PLC:** Phospholipase C

**IMP:** Inositol monophosphatase　　**IPK2:** Inositol 1,4,5- tris-phosphate kinase

**MIK:** Myo-inositol kinase　　**ITPKs:** Inositol (1,3,4) $P_3$ 5/6-kinases or inositol triphosphate kinases

**PtdS:** Phospatidyl inositol synthase　　**IPK1:** Inositol 1,3,4,5,6- pentakisphosphate-2-kinase

RNAi mediated gene silencing was reported

**Figure 4.** Rice phytic acid metabolism pathway. Adapted from Suzuki et al. [66]. RNAi-mediated gene silencing reported in the circled enzymes in this pathway [32,33,65,67].

The very first step of phytic acid biosynthesis in rice seed is catalyzation by myo-inositol-3-phosphate synthase (MIPS) enzyme. This enzyme was targeted for silencing in rice by using *CaMV35S* promoter by Feng and Yoshida [68], although seed-specific promoters such as Glutelin B-1 (GluB-1) and Oleosin 18 (Ole18) are preferred for maximum phytate accumulation in seed as was demonstrated by Kuwano et al. [65], Kuwano et al. [67], and Ali et al. [32]. After *MIPS* silencing, rice seeds showed change in myo-inositol level, as *MIPS* is a precursor for the de novo synthesis of myo-inositol. Therefore, enzymes involved at a later stage in phytic acid biosynthesis in rice should be targeted to reduce the phytate concentration in seeds without disturbing related important pathways.

Ali et al. [33] developed a Pusa Sugandhi II (PSII) indica rice cultivar by manipulating the expression of the final step key enzyme inositol-1,3,4,5,6-pentakisphosphate 2-kinase (IPK1) of phytic acid metabolism by silencing *IPK1* gene using Ole18 seed-specific promoter by RNAi technology. A 3.85-fold down-regulation in *IPK1* transcripts was observed for the transgenic seeds, which correlated to a significant reduction in phytate levels and increase in the amount of inorganic phosphate (Pi) and accumulated 1.8-fold more iron in the endosperm without hampering the growth and development of transgenic rice plants.

Karmakar et al. reported phytic acid downregulation of 46.2% by seed-specific RNAi-mediated gene silencing of an inositol triphosphate kinases (*ITPK*) homolog (*OsITP/6K-1*) in Khitish indica rice variety and found an 1.3-fold increment of iron accumulation in seed with 1.6-fold zinc and 3.2-fold bioavailability of Pi [31].

In terms of the long term effect of RNAi-mediated silencing of the phytic acid pathway, ferritin overexpression in rice plants for development of 'high iron' rice has been studied via phenotypic and agronomic performance data under the facilities of the University of Calcutta, India (Figure 5a). GM rice plants were grown for multiple generations to establish homozygous plant lines, and iron accumulation in seed and seed morphology were studied. No alteration of seed structure in high iron rice was reported (Figure 5b).

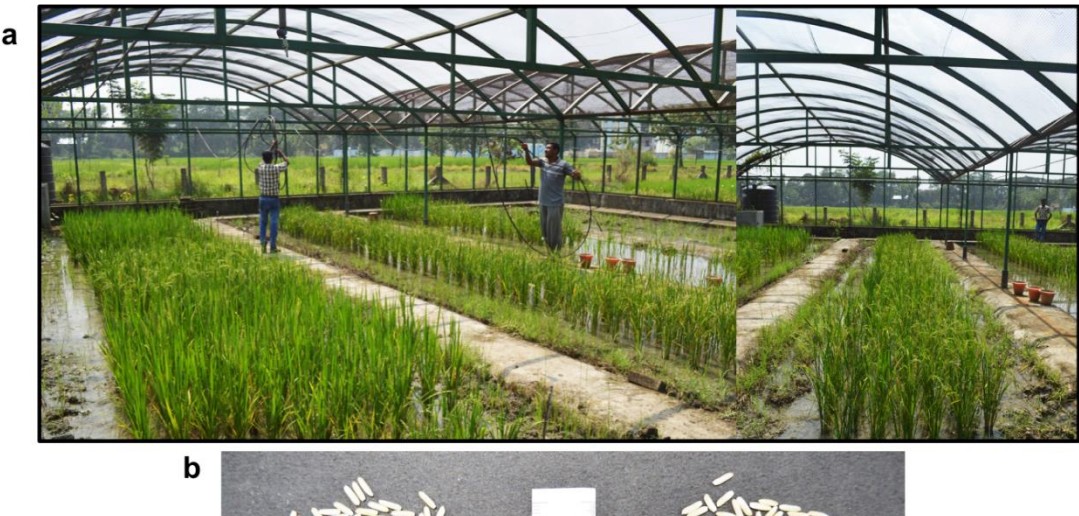

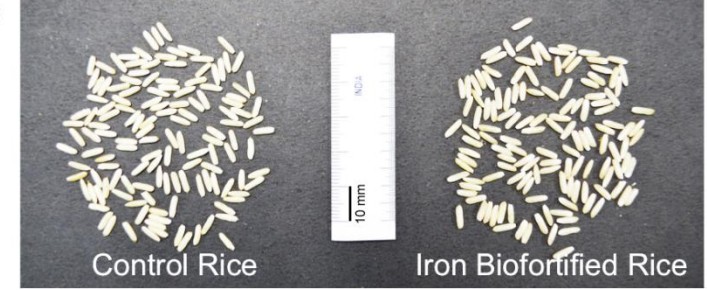

**Figure 5.** Field performance testing of (**a**) genetically modified (GM) biofortified high iron rice developed under public laboratory facility of the University of Calcutta, India. (**b**) No change in seed grain morphology found between control rice (non-GM) and biofortified rice (high iron rice) developed by using RNAi-mediated gene silencing technology [32,33] or overexpression of the *ferritin* gene in it [13,14]. Bar represents 10 mm.

### 3.2.8. Release of Phytic Acid Bound Iron

As an alternative to the silencing of phytic acid metabolism genes, expression of fungal (i.e., *Aspergillus fumigatus*) phytases enzyme in rice is a promising biofortification strategy. Phytase can catalyze the hydrolysis of phytic acid (phytate) releasing the chelated minerals (i.e., $Fe^{2+}$, $Zn^{2+}$, $Ca^{2+}$, and $Mg^{2+}$) including phosphate, resulting in a greater mineral bioavailability [69]. *A. fumigatus phytase* (*Afphytase*) has a thermotolerant property which makes it more suitable for food processing and biofortification of staple crop-related applications [70]. The *Afphytase* gene was introduced into rice by Wirth [35], Boonyaves et al. [36], and Boonyaves et al. [40], and the resulting GM rice showed increased iron accumulation in polished grain. Overexpression of *appA* (phytase) gene from *Escherichia coli* in Khitish (indica) rice cultivar enhanced twofold iron and threefold zinc accumulation in rice seed with a fourfold increase of inorganic phosphorus (Pi) level [37]. Lucca et al. introduced *Afphytase* into the rice endosperm with a rice cysteine-rich metallothionein-like protein (for enhancing of iron absorption) [34]. Cysteine facilitates non-haem iron absorption [71] and each metallothionein (MT) molecule is reported to contain large amounts of cysteine (20 of 60 amino acids in mammalian MTs [72] and 12 of 74 in plant MTs [73]). Through endosperm-specific overexpressing of MT in rice, the cysteine concentration of seed protein can be increased in grains, which would lead to enhancement of iron bioavailability. In the developed GM Taipei-309 rice, the phytase level was 130-fold increased (in the grains) and in a test (simulated digestion) it showed complete degradation of phytic acid.

### 3.2.9. Combination of Multiple Transgenes

Stacking of multiple transgenes in different combinations was applied to achieve GM iron biofortified rice. Wirth et al. developed GM Taipei-309 rice with 6.3-fold increased iron accumulation in polished grain by introducing *Pvferritin*, *AtNAS1*, and *Afphytase* genes [35]. Masuda et al. developed GM Tsukino Hikari rice through introduction of *SoyferH2*, *HvNAS1*, and *OsYSL2* genes and observed sixfold increased iron accumulation in brown rice grain [38]. Trijatmiko et al. also reported sixfold increase in iron accumulation in polished IR64 grain through *SoyferH1* and *OsNAS2* gene combination [41]. Aung et al. transferred *SoyferH2*, *HvNAS1*, and *OsYSL2* gene combination and achieved GM Paw San Yin rice variety with an iron increase of 3.4-fold in polished grain [42]. Boonyaves et al. found 4.7-fold iron increase in polished grain of GM Nipponbare having *AtIRT1* (encode for $Fe^{2+}$ transporter), *Pvferritin*, and *AtNAS1* gene combination [40]. A combination of four genes (*AtIRT1*, *Pvferritin*, *AtNAS1*, and *Afphytase*) has been transferred in Taipei-309, resulting in 4.3-fold iron increase in GM rice polished grain [36]. Masuda et al. introduced three barley genes (*HvNAS1*, *HvNAAT-A*, *HvNAAT-B*) that participated in mugineic acid biosynthesis with soybean *ferritin* (*SoyferH2*) gene in Tsukino Hikari rice and found fourfold increase in iron accumulation in polished grain [39]. These GM rice explain that a perfect combination of transgenes could increase iron concentration in polished rice and in future the increase could be more than sixfold.

## 4. High Zinc Rice

Zinc is essential for regulating absorption of Fe in the intestine, and sufficient quantity of zinc (along with iron) is crucial for treating IDA [74]. Zinc is also vital for physical growth and development, functioning of the immune system, reproductive health, sensory functions, and neurobehavioral development. Most importantly, zinc is required for the activation of over 300 enzymes and proteins (i.e., zinc finger proteins) as it is the only metal to be involved in all six classes of enzyme structure and function [75]. Zinc is essential for proper functioning of many transcription factors (regulators), zinc finger proteins, and many enzymes that require Zn in different forms. About 17.3% of the world's population is zinc deficient and more than 400 million children (under the age of five) die every year due to zinc deficiency [76]. Zinc deficiency causes serious adverse health effects in children (varying with age)—diarrhea, low weight gain, stunting growth, and anorexia are common in children. Neurobehavioral changes are observable in infants, whereas changes in skin and dwarfing are more

frequent in toddlers and school children [77]. Rice is the main source of zinc intake in Asian countries such as Bangladesh, where rice alone provides 49% of dietary zinc to children and 69% of dietary zinc to women [78].

In 2013, CGIAR-HarvestPlus released a zinc biofortified rice variety developed through conventional breeding in Bangladesh. Currently about 1.5 million farming households accepted eight varieties of zinc-biofortified rice and have since been growing them [79]. The Indian Institute of Rice Research, Hyderabad, developed a biofortified semi-dwarf, medium duration (125 days) variety with a non-lodging plant type named IET 23832 (DRR Dhan 45) with a zinc concentration of 22.6–24.00 ppm in polished grain (https://icar.org.in/node/6293, accessed on 14 April 2019). The IET 23832 was also developed by conventional breeding by using HarvestPlus material with some important qualities such as desirable amylose content (20.7%), ensuring good cooking quality, as well as resistance (moderately) to rice blast disease (*Magnaporthe grisea*), sheath rot disease (*Sarocladium oryzae*), and rice tungro virus infection.

The molecular breeding strategies involved in zinc biofortification are similar to that of iron biofortification. Furthermore, the uptake and homeostasis of zinc and iron are closely linked in cereals. In rice, iron and zinc uptake is mediated by members of the zinc and iron-regulated transporter protein (ZIP) family. Several ZIP family proteins are present in rice—*OsIRT1* and *OsIRT2* are ferrous iron transporters [53]. *OsZIP1*, *OsZIP2*, *OsZIP3*, and *OsZIP4* are associated with metal uptake and zinc homeostasis [80,81], and *OsZIP7a* and *OsZIP8* might encode an iron and zinc transporter, respectively [82]. The *OsZIP1* gene was upregulated under zinc-deficient conditions, whereas *OsZIP3* was upregulated under both conditions (zinc available and deficient condition) in rice [83].

Overexpression of *OsIRT* and *MxIRT* genes in GM rice resulted in increased iron and zinc concentration in rice grains [84,85]. Boonyaves et al. expressed *AtIRT1* with *Pvferritin* and *AtNAS1* genes for iron biofortification, and as a result they found a 4.7-fold iron increase increase in zinc also [40]. A combination of four genes (*AtIRT1*, *Pvferritin*, *AtNAS1*, and *Afphytase*) has been transferred in rice, and as a result iron and zinc accumulation in GM rice increased in polished grain [36]. Several studies have shown that the overexpression of *OsNAS* genes improved the iron and zinc concentrations by several folds in rice grain [21,22]. Overexpression of rice *OsNAS1*, *OsNAS2*, and *OsNAS3* done by Johnson et al. resulted in a twofold increase of zinc concentration in rice seed [21].

Ali et al. reported that RNAi-mediated silencing of *MIPS* gene of phytic acid metabolism pathway increased zinc, calcium, and magnesium concentration in milled rice grain along with iron [32]. A similar result of increment in metal concentration including zinc was found when another gene of phytic acid metabolism, *IPK1*, was silenced by Ali et al. [33] and *ITPK* (*OsITP/6K-1*) was silenced by Karmakar et al. [31]. RNAi-mediated silencing of negative regulators of iron in rice (i.e., OsHRZ2) also increased zinc concentration along with iron in grain when compared to control rice [29]. Zinc accumulation with iron in GM ferritin rice grain is quite common. The *ferritin* gene (*Osfer2*) was overexpressed in PSII rice, and accumulation of 2.09-fold and 1.37-fold of iron and zinc, respectively, was reported by Paul et al. [20].

## 5. Provitamin-A (β-Carotene) Biofortified Rice—'Golden Rice'

Dietary carotenoids have various health benefits such as decreasing the risk of eye disease and cancer. A number of carotenoids have been studied that aid in human health, cell differentiation, synthesis of glycoprotein, growth and development of bones, and have antioxidant properties and nutritional benefits, such as β-carotene, lycopene, lutein, and zeaxanthin [3,86]. β-Carotene may have added benefits as it can convert to vitamin-A. As mentioned before, brown rice is incapable of producing vitamin-A. Vitamin-A deficiency (VAD) is a worldwide phenomenon that affects Southeast Asia and Sub-Saharan Africa the most. According to WHO, half of the world's children suffering from VAD belong to these regions [3]. Development of carotenoid biofortified rice could be a solution for VAD, as carotenoids would be made available in polished rice grains [87,88]. Carotenoid biofortified rice is popularly known as golden rice (GR) because of its grain color. The yellowish orange color of

grains developed due to the seed-specific introduction of carotenoid biosynthesis pathway in rice and accumulation of β-carotene.

## 5.1. Development of Golden Rice

GR is an example of successful metabolic engineering of seed-specific carotenoid biosynthesis pathway in rice (Figure 6a). The initiation of metabolic engineering occurred with the introduction of daffodil (*Narcissus pseudonarcissus*) phytoene synthase (*PSY*) gene under endosperm-specific promoter to produce transgenic japonica rice (Taipei-309). Developed GM rice accumulates phytoene in seed, which is a key intermediate of provitamin-A [89]. This daffodil *PSY* gene in combination with bacterial (*Erwinia uredovora*) phytoene desaturase (*CRTI*) gene was introduced under the control of the endosperm-specific glutelin promoter in the same japonica rice variety (Taipei-309) by *Agrobacterium*-mediated transformation [90]. This gene combination was able to produce 1.6 µg/g of total carotenoids in rice endosperm (grains). Similar expression of total carotenoids in rice grain (1.05 µg/g) by this gene combination was reported in indica rice (Figure 6b) [91]. Later, more popular indica varieties such as IR64 and BR29 were genetically engineered for seed carotenoid enhancement. Total carotenoids increased to 9.34 µg/g in such modified indica varieties and β-carotene (provitamin A) in polished IR64 seed was reported as 2.32 µg/g and for BR29 it was 3.92 µg/g (Figure 6c) [92]. The highest expression (37 µg/g) of total carotenoids was reported by Paine et al. by introducing maize *ZmPSY* and *Erwinia uredovora CRTI* genes under endosperm-specific promoter [93]. This GR was named as GR2 (golden rice-2). Bai et al. developed GR by expressing maize phytoene synthase (*ZmPSY1*), bacterial phytoene desaturase (*PaCRT1*), with the *Arabidopsis thaliana* genes *AtDXS* (encoding 1-deoxy-D-xylulose 5-phosphate synthase for continuous supply of metabolic precursors) and *AtOR* (the ORANGE gene for formation of a metabolic sink), which produced up to 31.78 µg/g total carotenoids in rice grain [94]. The sequential events of GR development are represented in Figure 7.

Parkhi et al. and Baisakh et al. demonstrated how a marker gene can be successfully removed from GR, thereby making it 'marker free' GR [95,96]. Marker genes—specifically antibiotic resistant genes such as *hpt* (hygromycin-B phosphotransferase against hygromycin-B), *nptII* (aminoglycoside phosphotransferase against kanamycin, neomycin, paromomycin), *bla* (beta lactamase enzyme against ampicillin), and *aad* (3″(9)-O-aminoglycoside adenylyltransferase enzyme against spectinomycin and streptomycin)—are commonly used in molecular breeding (transgenic plant development), but commercial food crops such as biofortified rice or other crops need to be free from such marker genes to address the regulatory process and consumer concern. Specific techniques and modified plant transformation approaches have been developed to remove such marker genes from transgenic crops. Techniques such as transposon-mediated elimination of marker gene [97], intrachromosomal homologous recombination [98], site-specific recombination Cre/LoxP [99], and FLP/FRT [100] systems have been developed. Modified transformation techniques such as co-transformation of marker genes and target genes (gene of interest) and excision of the marker gene in subsequent generations by genetic segregation are some methods for marker-free transgenic rice development [95,101]. A non-antibiotic selection system (Positech), where only transgenic plants housing the phosphomannose isomerase (*pmi*) gene can survive in mannose supplemented selection media, has been reported as an alternative to antibiotic marker-free GR development [92].

Datta et al. [92] pointed out a phenomenon during development of indica GR varieties (IR64 and BR29), where β-carotene in $T_2$ seeds increased when compared to $T_1$ seeds. This enhancement in expression level of β-carotene was considered as a positive post-transgenerational consequence of carotenoid biosynthesis in GR.

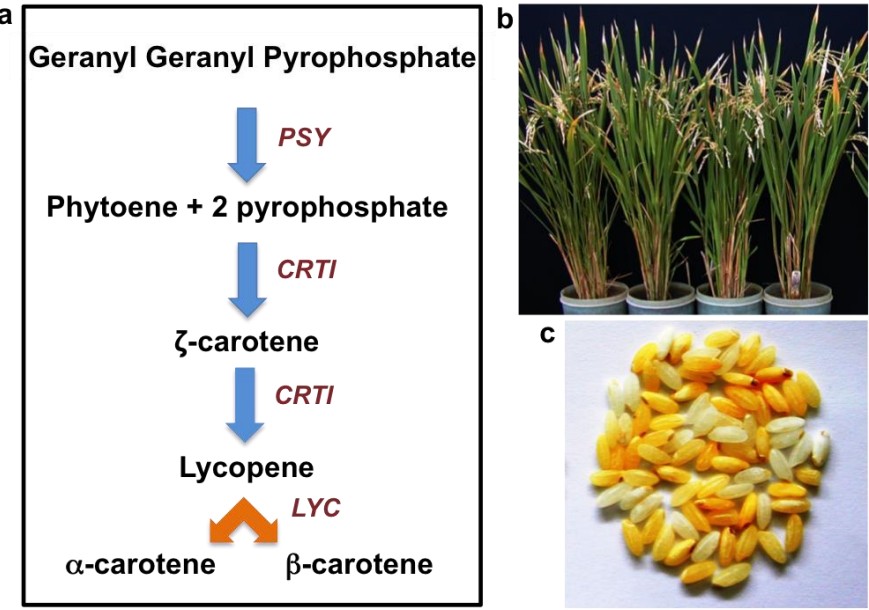

**Figure 6.** Metabolic engineering of rice to introduce carotenoid biosynthesis pathway to develop 'golden rice' (GR). (**a**) The phytoene synthase (*PSY*), phytoene desaturase (*CRTI*), and lycopene-beta-cyclase (*LYC*) genes have been introduced from other sources to rice [91]. (**b**) Initially GR was developed in japonica (Taipei-309) rice varieties and later in indica (BR29) GR [102]. (**c**) Mixture of indica GR (yellowish-orange color) and its control (white color) rice grains showing no structural difference between them but differences in color, owing to the content of β-carotene present in GR.

Different approaches of GR development have been reported, that is, development of introgress lines by breeding [96] and development through dihaploid homozygosity [88]. In dihaploid/double haploid (DH) homozygosity approach, GR plants were first developed by molecular breeding, then after successful anther or pollen culture from those plants, further multiplication was initiated. Thus rapid homozygous (isogenic lines) GR plants were achieved. This approach helped to achieve homozygosity in one or two generations, and avoided the extensive and time-consuming method of growing $T_0$ transgenic plants for multiple generations [88]. This approach of GR development can be applicable for other rice biofortifications for rapid variety development.

Detailed molecular characterization of GR [95], field performance analysis [102], and metabolic and proteomic analyses [103] have been performed in different laboratories worldwide, and have been found to be safe for human consumption [104]. Recently in 2018, three renowned international food safety regulatory agencies: Food Standards Australia New Zealand, Health Canada, and the United States Food and Drug Administration recommended GR for commercialization and gave positive feedback on it (www.irri.org/golden-rice, access on 20 April 2019).

### 5.2. Long Term Storage of 'Golden Rice'

Proper storage and maintenance of nutritional qualities of GR is a challenge. Seeds contain lipoxygenase (LOX) enzyme that catalyzes the insertion of molecular oxygen into PUFA (polyunsaturated fatty acids), yielding conjugated hydroperoxide products, which in turn oxidizes carotenoids and causes deterioration of seed nutritional quality [105]. The rice genome contains 14 types of LOX protein-encoding genes; among them the *r9-LOX1* gene is responsible for seed quality deterioration [106]. RNAi-mediated down-regulation of this *r9-LOX1* gene in GR under the control of *Oleosin-18* promoter improved the storage stability and viability of GR seeds [107]. This strategy could be useful for long term storage of rice seeds in future.

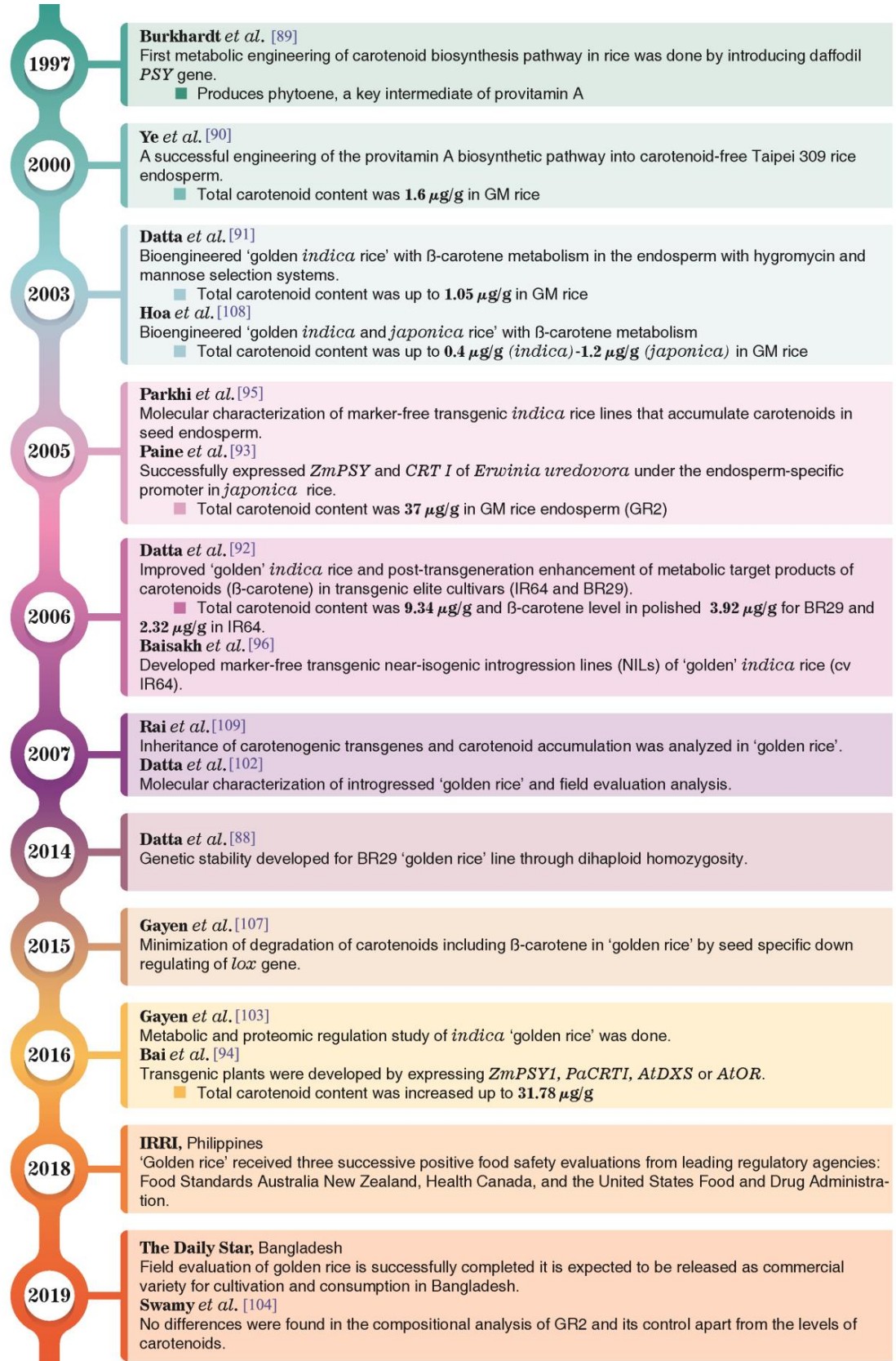

**Figure 7.** A timeline view of significant achievements in golden rice Research. Exclusive references for this infographic are given in [108,109].

## 6. Food Quality and Safety Analysis of Biofortified Rice

Safety and quality assessment of GM rice consistently acquires a worldwide research priority. Such assessment studies (including substantial equivalence studies) are not only important to understand seed and nutrition biology, but also facilitate rapid approval of GM rice for human consumption. A comparative analysis of nutritional compositions of GM 'high iron rice' IR68144 with its non-transgenic counterpart has been reported and found substantially equivalent to its counterpart, except in its increased amount of iron and zinc [110].

A recent multi-locational and multi-seasonal compositional study on GR (GRE2 rice) and its non-transgenic counterpart (PSBRc82) revealed no-significant difference among them other than the level of β-carotene and related carotenoids [104]. Similar findings were reported for indica GR variety BR29 (event name SKBR-244) and its non-transgenic control under a metabolic regulation study [103].

## 7. Regulatory Challenges of Biofortified Rice

The future of rice nutrition improvement largely depends on molecular breeding approaches, specifically iron, zinc, and provitamin-A fortification, as conventional breeding has a very limited scope. No high iron- and zinc-fortified rice (in polished grain) has been found from a screening of more than 20,000 rice varieties from Asia, Latin America, and the Caribbean [111], and conventional breeding has so far been unsuccessful in high iron polished rice development. Wild varieties of rice such as *Oryza rufipogon*, *Oryza nivara*, *Oryza latifolia*, and *Oryza officinalis* contain more Fe and Zn than cross-bred cultivars, but their low yield is a major limitation.

Rice is a major source of nutrients and contributes up to 70% of daily calories for more than half of the world's population [112]. In developing countries, the average person's diet is not free from micronutrient deficiency, and the availability of biofortified rice under such circumstances would be a sustainable solution. Rapid biofortified, biotic, and abiotic stress tolerant, high yielding rice variety development requires new molecular breeding technologies. Development of GM rice by transgenic approach is one such technology. As any new technology requires safety and security assessment—biofortified GM crops (rice) are no exception. Biofortified GM rice faces different regulatory challenges on the basis of country of concern. Most of the developing countries do not have necessary infrastructure, sophisticated laboratories for food quality and safety analysis, marketing strategies, or proper policies and political will. This leads to delay in reaping benefits of biofortified GM rice by its consumers. The regulatory faith of CRISPR/Cas9 (clustered regularly interspaced short palindromic repeats-associated endonuclease Cas9) genome edited crops have not been decided yet. The USDA (United States department of agriculture) has exempted the application of strict GMO (genetically modified organism) regulations in many CRISPR-edited crops, whereas the Court of Justice of the European Union has recently judged that it will be regulated as per GMO regulation guidelines [113].

After almost 20 years of GR development there is still no report of its commercialization. However, hope remains as the Philippines and Bangladesh are reaching the final stage of the regulatory system of GR [114]. Such delay of GR commercialization includes multiple factors such as the destruction of field trials by anti-GMO activists in 2013 in the Philippines, performance of lead GR event (GR2-R1) in the field compared to non-GM control, and some gene integration-related issues of GR2-R1 events (disrupted *OsAux1* gene and disrupting the transport of auxins) in subsequent breeding programs for the indica variety Swarna [115,116].

Amidst social, economical, and political regulations, "hidden hunger" continues to accompany the less fortunate. Meanwhile, crop scientists keep working to bring about change using new technologies, hoping that the malnourished people of the world will obtain proper nutrition in the near future.

## 8. Discussion

Rice biofortification via genetic engineering, based on transgenic and RNAi-mediated silencing of antinutrient pathways, led to the development of some rice varieties that showed 6.3-fold increase in iron [35], 2.0-fold increase in zinc [21], and 37 µg/g total carotenoid [93] in biofortified rice grain. As technological improvement of plant genetic modification is a dynamic process, increasing detailed knowledge of rice genome sequencing data will lead to more valuable biofortified rice in the future. Recently developed CRISPR/Cas9 technology has shown some reflection of it by changing rice seed qualities [117–119], plant growth [120], and rice leaf stomatal density [121]. Development of new biofortified rice events on the basis of GR, such as the aSTARice (or Astaxanthin Rice), have been initiated [122]. The aSTARice contains carotenoids and keto-carotenoids (such as astaxanthin and canthaxanthin) that are beneficial antioxidants. There is a continuous search in the gene pool of rice varieties and other organisms for candidate genes useful for rice biofortification projects to fight against hidden hunger. Descalsota et al. recently searched through the genome-wide association studies (GWAS) using a multi-parent advanced generation inter-cross (MAGIC) plus rice population to identify QTLs (quantitative trait locus) and SNP (single-nucleotide polymorphism) markers for biofortification [123]. Findings confirmed that iron and zinc homeostasis genes *OsMTP6*, *OsNAS3*, *OsMT2D*, *OsVIT1*, and *OsNRAMP7* were co-located with QTLs. This kind of knowledge could be beneficial for rice biofortification in the near future.

Combination of iron, zinc, and vitamin-A has a potential synergistic interaction in human health and increases bioavailability of these minerals. Zinc assists in the synthesis of vitamin-$A_1$ (retinol) binding protein and increases lymphatic absorption and transport of vitamin-A. Similarly, vitamin-A also affects zinc lymphatic absorption and transport via regulating zinc-dependent proteins [124]. Vitamin-A has quite similar effects on iron. Carotenoids enhance the transport of iron from the human gut to the mucosal cell membrane and increase iron bioavailability [125]. Synergy in absorption of iron and zinc in cells has been well established [126]. Such natural 'synergistic effect' motivates plant breeders to combine high iron, zinc, and vitamin-A traits in future to develop superior varieties of biofortified rice that can be considered as an improvement to the prevalent approaches. Recently, five Nipponbare rice lines (CP22, CP87, CP97, CP101, and CP105) have been developed for iron, zinc, and β-carotene biofortification by expressing *AtNAS1*, bean ferritin (*Pvferritin*), bacterial *CRTI*, and *ZmPSY* in a single genetic locus [127]. $T_3$ plant progeny gave results of 1.57 to 2.69 µg/g DW β-carotene, 1.2- to 1.5-fold increment of iron, and 1.1- to 1.2-fold increment of zinc concentration in polished grains as compared to non-transgenic controls. This rice biofortification approach can be considered an effective method to address multiple micronutrient deficiencies at once.

Along with finding the solution to the major deficiencies related to iron, zinc, and vitamin-A, rice biofortification can also be targeted to other minerals and vitamins to fight against hidden hunger. Rice biofortification for folic acid (or folate), thiamin, riboflavin, niacin, pantothenic acid, pyridoxine, biotin, vitamin-B12, ascorbic acid, vitamin-D, and vitamin-E are in steady progress [3].

As rice biofortification is a sustainable approach to fight against "hidden hunger" over chemical food supplements, it should be a priority for research or technological advancement studies for affected countries. As the techniques are directly beneficial to rice consumers of affected nations, biofortified rice developers, policy makers, stakeholders, and philanthropists should focus on policies such as the PPP model (public private partnership) in Agri-Biotech research, 'freedom to operate' (FTO) biofortified rice varieties developed by private companies, area-specific production, better storage facilities, international rice distribution policies, and developing awareness on the nutritional value of other locally available foods.

**Author Contributions:** Conceptualization, S.K.D.; writing—original draft preparation, S.M.; writing—review and editing, K.D. and S.K.D.

**Funding:** DBT and ICAR (Indian Council of Agricultural Research), Government of India for financial support of research projects on rice.

**Acknowledgments:** 'Distinguished Biotechnology Research Professor Award' by the Department of Biotechnology (DBT), Government of India to S.K.D.

**Conflicts of Interest:** The authors declare no conflict of interest.

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
