# Peer review of "Rice Biofortification: High Iron, Zinc, and Vitamin-A to Fight against “Hidden Hunger”"

_agronomy, doi:10.3390/agronomy9120803_

Round 1
Reviewer 1 Report
The manuscript by Majumder et al. (agronomy-618302) is a concise review for biofortification of three major nutritional deficiencies of humans: iron, zinc and vitamin-A, in rice grains. Although some of the issues has been often reviewed elsewhere (i.g. iron biofortification), this review summarizes the three biofortification targets and their application issue, which would be beneficial for many readers. I would like to suggest revisions as follows:
1. Throughout the manuscript, the term “content” is mistakenly used for “concentration”, which should be thoroughly reconsidered.
2. The data for loss of nutrients during milling process in Fig. 1 and lines 56, 79 etc. are based on single reference (Ref. 5, which also appears to derive from another reference), and may not reflect actual range and variation. For example, Masuda et al. 2009 (Ref. 34) reports average Fe concentrations (NOT content) of ~19 ppm in brown seeds and ~4 ppm in polished seeds, indicating much more decreasing rate of Fe during polishing in wild-type rice cultivar Tsukinohikari. Such variation ranges should be reconsidered.
3. The authors miss another powerful strategy for Fe and Zn fortification: manipulation of regulators for Fe and Zn uptake and translocation. Specifically, knockdown rice lines of OsHRZ1 and OsHRZ2 ubiquitin ligases (negative regulators of Fe deficiency responses) efficiently accumulate Fe and Zn in grains under various conditions (Kobayashi et al. 2013 Nat. Commun). Also, overexpression rice lines of OsIRO2 transcription factor (positive regulator of Fe deficiency responses) accumulate Fe and Zn in grains when cultivated in calcareous soil pots. These points should be included in Table 1, section 3.2 and section 4.
4. Other points as listed below should be revised:
Line 77. The term “most essential” should be reconsidered, because there is no order of essentiality.
Line 78-79. In which part does this rice contain “38 ppm of iron”? Please specify.
Fig. 2. Unit of the vertical axis “gm/L” is unusual and should be revised.
Line 116-117. This description is not correct and misleading. Endogenous SoyferH1 and SoyferH2 are not controlled by “endosperm specific promoters”. This should be revised with appropriate original reference(s). Ref. 16 uses SoyferH1 and SoyferH2 as transgenes using “endosperm specific promoters”.
Line 135. Nicotianamine is not a kind of PSs that are secreted by plants for Fe acquisition, but it is their precursor.
Line 138. The “release” of PSs requires transporters such as TOM1 in rice (Nozoye et al. 2011 J. Biol. Chem.). Enzymes cannot “release” them.
Line 162-170. Refs. 39, 28 and 30 are not appropriate here. Original references should be cited instead.
Line 208. “CaMV35S” should be CaMV35S promoter.
Line 211. MIPS should not be “the only precursor” because it is an enzyme.
Line 235 and Fig. 5. Please clarify “the white scale”, or otherwise replace it to a scale bar.
Line 247. How does a rice metallothionein-like protein contribute to enhancing iron absorption? Please explain.
Lines 255-256. “Trijatmiko et al.” do not match with the ref. 61.
Lines 271-272. Zn-fingers are most well known as components of regulators such as transcription factors, while many enzymes require Zn as other forms. The description should be revised based on these aspects.
Line 342. Please explain “AtDXS and AtOR”.
Line 346. Please explain how to produce “marker-free” GR.
Line 348. Please explain “dihaploid homozygosity”.
Lines 312, 364, 407 and elsewhere. Grammatical errors should be corrected.
Fig. 7. 2016. “post-transgeneration” should be explained.
Lines 433-437. Actual achievements of ref. 110 should be described.
Author Response
Dear Sir,
You suggestions and queries are highly acknowledged, it helped to improve the quality of this manuscript and make it clearer to the readers. All corrections (RED COLOUR) as per your suggestions have been incorporated in the revised version of the manuscript. Correction on language has been marked as GREEN COLOUR in the revised manuscript. Please find attached a copy of our point wise response to your review report.

Reviewer 2 Report
Majumder et al. reported the recent progress on the studies of rice biofortification to increase grain iron, zinc, and vitamin-A levels. The manuscript is well written and the topic is important. Thus, I think it is a paper of strong interest to the readers of Agronomy. I have a couple of suggestions, that need to be modified before publication. Most of them are pretty straight forward.
Minor edits
L.11 “Oryza sativa” needs to be in italic
L.53 & L54 “polished (milled) rice” and “milled (polished) rice” are redundant.
L. 67. Remove one period.
L. 79. --- reduced to 8.8 ppm in brown rice after ----4.1ppm in milled rice. Refer Figure 1 here.
Figure 1. Grain iron concentration in rough rice should be 8.8 ppm as explained in the text (i.e. L.56). Source of information should also be referred in the figure legend.
List of references needs to be carefully corrected and some recent works may still be referred.
Author Response

(The authors gave the same response as above.)
